# Accurate and Reliable Classification of Unstructured Reports on Their Diagnostic Goal Using BERT Models

**DOI:** 10.3390/diagnostics13071251

**Published:** 2023-03-27

**Authors:** Max Tigo Rietberg, Van Bach Nguyen, Jeroen Geerdink, Onno Vijlbrief, Christin Seifert

**Affiliations:** 1Faculty of EEMCS, University of Twente, 7500 AE Enschede, The Netherlands; 2Institute for Artificial Intelligence in Medicine, University of Duisburg-Essen, 45131 Essen, Germany; 3Hospital Group Twente (ZGT), 7555 DL Hengelo, The Netherlands

**Keywords:** natural language processing, health informatics, BERT, text classification

## Abstract

Understanding the diagnostic goal of medical reports is valuable information for understanding patient flows. This work focuses on extracting the reason for taking an MRI scan of Multiple Sclerosis (MS) patients using the attached free-form reports: Diagnosis, Progression or Monitoring. We investigate the performance of domain-dependent and general state-of-the-art language models and their alignment with domain expertise. To this end, eXplainable Artificial Intelligence (XAI) techniques are used to acquire insight into the inner workings of the model, which are verified on their trustworthiness. The verified XAI explanations are then compared with explanations from a domain expert, to indirectly determine the reliability of the model. BERTje, a Dutch Bidirectional Encoder Representations from Transformers (BERT) model, outperforms RobBERT and MedRoBERTa.nl in both accuracy and reliability. The latter model (MedRoBERTa.nl) is a domain-specific model, while BERTje is a generic model, showing that domain-specific models are not always superior. Our validation of BERTje in a small prospective study shows promising results for the potential uptake of the model in a practical setting.

## 1. Introduction

With the rising costs for healthcare [1], policy makers want the reduce costs while ensuring proper patient treatment. To inform policy makers where and how improvements can be made, it is key that patient flows are quantified. Knowing where to go is impossible without knowing where one stands. Furthermore, with the same data (quantified patient flows) it is possible to train models to, e.g., predict risk probabilities or advise in treatments, enabling further healthcare improvements. Clearly, well-annotated data is important. However, medical records are often free-form reports, hiding information in unstructured sentences and inconsistent language. Requiring medical personnel to manually enter information into dedicated discrete fields, instead of writing free-form reports (to bypass the natural language problems), is more time-consuming, inflexible, and error-prone [2,3,4,5]. Therefore, to (further) implement Data-Driven Healthcare—the latest development in healthcare [6]—one needs automatic analysis of natural language. One example is extracting the reason for taking an MRI scan of Multiple Sclerosis (MS) patients using the attached free-form reports: Diagnosis, Progression or Monitoring.

Automatic analysis of free-form electronic health records is nearly impossible without specialized Natural Language Processing (NLP) models. Specialized models are needed because general language models are (A) often only able to handle English corpora and/or (B) only able to handle ’standard’ language as found in books or on the web. Namely, medical records differ considerably in their language usage. They contain a magnitude of medical terms: e.g., symptoms, names of diseases and treatments, devices, and instruments [7], which occur only seldom in books, news articles and on the web (which are frequently used for training NLP models). Furthermore, the records are often written in a time-efficient manner, resulting in—amongst others—unintentional grammatical and spelling mistakes, and intentionally omitting function words and using domain-specific abbreviations.

If a model is able to correctly classify the reports, it is also key that the model classifies the reports based on the right reasons instead of spurious features. This can be investigated through the usage of eXplainable Artificial Intelligence (XAI) techniques, which give insight into the inner workings of the model. The goal is to have a model of this domain that is accurate, is able to correctly classify the reports reliably, and uses the correct features for its decision.

In this study, we apply multiple state-of-the-art NLP models to classify the MRI reports of MS patients. MRI scans are conducted for specific purposes on MS patients, radiologists produce a report for each scan that details the underlying rationale for performing the MRI. Using NLP models, we extract the reasons for taking the MRI, quantifying the model’s ability to classify MRI reports from MS patients. It’s worth noting that we do not classify the MRI scans directly, but rather the accompanying reports. We compare conventional NLP models against Bidirectional Encoder Representations from Transformers (BERT) models. BERT is a machine learning framework for NLP introduced by Devlin et al., composed of Transformer encoder layers and trained to learn latent representations of words and sentences [8]. Next, we investigate their ability through the usage of XAI techniques and compare the explanations with explanations given by a domain expert (a board-certified radiologist specializing in neurology). We show that BERTje outperforms other BERT models in this task, both by achieving high accuracy and reasoning close to a domain expert. Most notably, BERTje outperforms a domain-specific BERT model, indicating that for this task a domain-generic model is superior. Additional to the retrospective evaluation, we validate the model in a small prospective study and observe similar performance on the new dataset.

## 2. Related Work

We review work on natural language processing, and the BERT transformer model with special attention to the Dutch language. Further, we briefly methods from the area of XAI aiming at eliciting the reasoning of the model.

### 2.1. Natural Language Processing and Language Models

In classical natural language processing, the text is converted into vectors by first creating a vocabulary of words and then estimating the importance of a term for a document. A prominent example for estimating the importance of a term is Term Frequency-Inverse Document Frequency (TF-IDF) [9]. The most recent revolution in NLP came with the introduction of a new neural network architecture called Transformers [10], originally introduced for machine translation, which utilizes attention layers and is able to handle long-range dependencies in a sentence. This architecture has been used in auto-regressive models such as GPT [11], GPT-2 [12] and XLNet [13]. To counter the unidirectional constraints of GPT and GPT-2, a bidirectional Transformer model, Bidirectional Encoder Representations from Transformers (BERT) [8], was introduced.

BERT has shown potential in a wide range of fields. Originally created and published in 2018, BERT was designed to learn deep bidirectional representations from unlabelled text [8]. It is pre-trained using two unsupervised tasks: Masked Language Modelling (MLM) and Next Sentence Prediction (NSP). BERT’s primary limitations are the significant model size which makes it computationally demanding to train and apply, as well as its incapability to handle non-English corpora. While BERT was created for tasks with an English corpus, mBERT was created using all Wikipedia pages with 104 different languages, making it able to handle tasks in multiple languages [8]. However, this resulted in worse or comparable performance to monolingual models with BERT’s architecture settings [14]. It has been speculated that this is because Wikipedia is not representative of general language use [15]. Currently, there exist monolingual models in at least 21 different languages [16]. For Dutch, there are three different monolingual domain-generic models.

### 2.2. Dutch BERT Models

The three Dutch domain-generic models are BERT-NL [17], BERTje [15], and RobBERT [18]. BERT-NL [17] is a BERT-based model that was pre-trained exclusively on the SoNaR-500 corpus [19], a corpus comprising 500 million words from a wide range of domains and genres. In contrast, BERTje (also BERT-based) was not only pre-trained on this corpus but also on books, news articles and related documents, and Wikipedia articles (total 12 GB) [15]. The last model is RobBERT [18] (RoBERTa-based), which was pre-trained on the Dutch part of the OSCAR corpus (39 GB), a pre-processed version of the 2018 Common Crawl [20]. RobBERTv2 outperformed BERTje in Die/Dat Disambiguation (DDD) and Sentiment Analysis (SA) [18]. However, a larger pre-training dataset does not exclusively mean better performance. BERTje has shown better results than RobBERT (but also BERT-NL and mBERT) in Named Entity Recognition (NER) and Part-of-speech tagging (POS) [21]. Additionally, a distilled version of RobBERT has been created to increase efficiency with only a small decrease in performance [22].

BERT models can be customized for specific domains by training them further with domain-specific vocabulary. There are three types of BERT models: (i) pre-trained on a generic corpus; (ii) pre-trained on a generic corpus and further on a domain corpus; and (iii) pre-trained exclusively on a domain corpus. All types require fine-tuning on a specific task after pre-training. Domain-specific BERT models can have significant benefits [23,24], but training from scratch requires a large amount of data and computational resources. Examples of Dutch domain-specific models include ArcheoBERTje [25] (type ii), Legal-RobBERT [26] (type ii) and MedRoBERTa.nl [27] (type iii).

### 2.3. Explainable AI

As AI technology advances, understanding the inner workings of models becomes increasingly important. XAI techniques can be used to verify the trustworthiness of models and gain insight into how they can be improved. In NLP, there are several options for explaining the inner workings of models, which can be categorized based on their approaches, such as surrogate model, example-driven, induction, provenance, and feature-importance [28]. Surrogate model XAI techniques involve training a second model to explain predictions, while example-driven XAI techniques provide comparable examples to explain predictions. Induction XAI techniques use human-readable representations, and provenance XAI techniques show a subset of the prediction process [29]. Feature-importance XAI techniques, such as Integrated Gradients (IG) [30], Local Interpretable Model-Agnostic Explanation (LIME) [31], and Shapley Additive Explanation (SHAP) [32], quantify the impact of specific features on the model’s predictions.

### 2.4. Natural Language Processing on Radiology Reports

The usage of NLP for the automatic analysis of radiology reports has a long history. In recent history, it is mostly used for information extraction and report classification. Examples of the former are Named Entity Recognition in brain MRI and CT reports [33], and quantifying number, associated factors, and topography of cerebral microbleeds from brain MRI report [34]. Examples of the latter are the classification of brain MRI reports into acute ischemic stroke [35], classification of ischemic stroke subtype [36], and systematically identification silent brain infarction cases [37]. BERT-based methods are also used in this domain, e.g., in labeling radiographic images, so they can be used as training data for a further model [38,39]. Specifically for MS, rule-based NLP algorithms were developed to identify and extract information concerning the clinical course of disease [40], feature-extraction with Naïve Bayes classification was used to diagnose patients with MS [41], and a Convolutional Neural Network was used to predict the Expanded Disability Status Scale (EDSS) to monitor the progression of MS patients [41]. More relevant to this paper, BERT was also used for MS. Namely, MS-BERT was created to generate embeddings and predict EDSS together with MSBC (a classifier for applying MS-BERT) [42]. MS-BERT has been trained on notes from neurological examinations. However, to our knowledge, BERT-based models have never before been used on Dutch radiology reports from MS patients.

## 3. Dataset and Preprocessing

We use a dataset of 826 reports for MRI scans on 360 different Multiple Sclerosis (MS) patients (76.1% female, 24.9% male). The average age of a patient at the time of scanning was 44 ± 13 years. 22.5% came from Almelo and 14.2% from Hengelo (two municipalities where the hospital has a location) and the other 60.3% from surrounding municipalities in the province (Overijssel). 3.1% came from municipalities outside the province.

In each report, the radiologist notes relevant information about the patient (symptoms, medical history), their findings and a conclusion in a report. The reports are from MRI scans with 3DT1 after contrast (T1 Vibe FS contrast) and/or 3DFLAIR (T2 SPACE Dark fluid), made on patients under the Dutch MS invoice code (0330-0531 or 0330-9920). The dataset contains reports from over 5 years (1 January 2017 up to and including 3 May 2022).

Each report was manually classified by a radiologist into one of three classes: (1) Diagnosis; (2) Progression; or (3) Monitoring. This classification was performed in accordance with the MAGNIMS (Magnetic Resonance Imaging in MS) criteria [43]. Table 1 provides an overview of the dataset, the number of reports is calculated after the data cleaning steps detailed later in this section. The dataset is quite imbalanced, therefore we applied to oversample before training the models (cf. Section 4.3 of details). Furthermore, class 3 can semantically be seen as a subset of class 2. Since there is no overlap between classes 1 and 2, we treat the problem as a single-label classification problem instead of a multi-label classification problem.

We apply the following standard preprocessing steps: tokenization, lemmatization, and lower casing. We remove new line characters, stopwords (using the list provided in the NLTK toolkit), punctuation, single character-words, and numbers. Tokenization and lemmatization is not perfect (in Dutch), especially for medical words. Examples of incorrect processed words are abbreviations: “afwk” is short for “afwijking” (impairment) and should be processed to “afwijken” (impair) but is unchanged, domain-specific words: “susceptibiliteitsartefacten” (susceptibility artifacts) should be processed to “susceptibiliteitsartefact” but remained unchanged, and spelling errors: “coordinatie” should be corrected to “coördinatie” (coordination) but remained unchanged. These examples can be found in context in the reports provided in Appendix A.

To verify the intra-annotator agreement, we ask the radiologist to re-classify a subset of 51 reports. To reduce manual labeling effort, we chose the reports for relabelling as follows. We randomly selected 8 reports of each class (i.e., 24 in total), and 22 reports that were consistently classified wrong by a preliminary version of a model. We added 5 reports (1 for class 1, 1 for class 2, 3 for class 3) that that do not contain words identified as relevant for their ground-truth class (using feature importance and preliminary versions of the models).

Next, the radiologist reclassified the reports without access to the initial label. If a report is classified differently than the first time, we ask the radiologist to classify those again so we can take the majority as ground truth. Analyzing the annotations, we observe the following error cases:**Ambiguity:** Both, classes 1 and 2 are correct (2 reports).**Exception:** This report is an exception (3 reports). Patient is pregnant, patient switched hospital, or standard protocol was not followed.**Exclusion:** This report should not be in the dataset. The patient is not a MS patient, or the report is not about an MRI scan related to their MS (3 reports).

The ambiguous cases and exceptions were among the 22 reports that were constantly mislabelled by preliminary versions of the classifier. All reports in Exclusion belong to class 1 (Diagnosis). Overall, 13 out of 51 reports were labeled differently than the first time. All of them were constantly mislabelled by the classifiers before. Based on the subset (without the exclusion cases), this implies a Cohen’s kappa score of 0.6. However, when only considering the randomly selected reports, the kappa score is 1, implying a high intra-annotator agreement. The label of these reports was corrected and the reports in Exclusion were removed before training. The numbers given in Table 1 reflect these changes, which results in 823 reports in total.

## 4. Experimental Setup

Our goal is to investigate whether NLP models can accurately classify MS reports and which models are more reliable for the task. We apply XAI techniques to acquire explanations for the model’s results, and then match these explanations with explanations given by a domain expert to check the model’s reliability. We repeat this for several models and compare the results. An overview of our approach is shown in Figure 1.

### 4.1. Machine Learning Models

We compare the deep learning models with classical approaches. Our baselines use TF-IDF feature space weighting or doc2vec embeddings. As classifiers, we use a Support Vector Machine (SVM), a Random Forest (RF) model and a logistic regression (LR) classifier with standard settings (https://scikit-learn.org, accessed on 10 October 2022). Both TF-IDF and doc2vec models are tuned using the training dataset. The doc2vec model is trained for 20 epochs with default settings [44]. Additionally, we report the performance of a majority classifier (i.e., assigning all test samples to class 1).

Further, we evaluate three language models based on the BERT architecture: BERTje [15], RobBERT [18], and MedRoBERTa.nl [27]. RobBERT v2 is used: https://huggingface.co/pdelobelle/robbert-v2-dutch-base (accessed on 10 October 2022). We use pretrained versions of all language models in two different settings, as feature representations or in an end-to-end fashion. To use the feature representations, we pass the reports through the pre-trained models and obtain the text embedding. We then train SVM, RF and LR classifiers on those embeddings. In the end-to-end approach, we do not use the hidden states as fixed features but train the model end-to-end, i.e., all model parameters are trainable. Specifically, we use the Adam optimizer with a learning rate of $5\times 10^{-5}$, and cross-entropy loss and train the models for 2 epochs [45]. We use the Keras deep learning library with Tensorflow models.

### 4.2. Model Explanations

We use three XAI feature importance methods to get insight into end-to-end BERT models. More specifically, we apply SHAP [32], LIME [31] and IG [30]. We apply SHAP, LIME and IG, on all test reports resulting in five important words per report with associated importance scores The methods retrieve the five most important words for the model prediction, independent of whether the prediction is correct or not. The goal is to explain the model as is, not the prediction as it should be were the model correct for the error cases.

We test the resulting explanations by measuring the classifier’s prediction with only (Fidelity) and without the important words (Deletion Check) [46]. If relevant words for the classifier were selected by the explanation method, one would expect that the accuracy stayed similar to the original model with only the important words and is significantly reduced if the important words are removed.

### 4.3. Evaluation and Verification with Domain Knowledge

To technically evaluate the models, we apply a leave-one-out evaluation protocol. In each training iteration, we use oversampling to account for class imbalance. We report accuracy, macro-averaged precision, recall and F1. Accuracy, precision, recall and F1 can be defined in terms of the number of true positives (TP), true negatives (TN), false positives (FP) and false negatives (FN). Accuracy is the ratio of true positives and all predictions. Precision is the ratio of TP and all samples that were predicted as true. Recall is the ratio of TP and all samples that should have been predicted as true. F1 is the harmonic mean of precision and recall:$$\begin{array}{cc} \mathrm{accuracy} & =\frac{\mathrm{TP}}{\mathrm{TP}\;+\;\mathrm{FP}\;+\;\mathrm{TN}\;+\;\mathrm{FN}}\\ \mathrm{precision} & =\frac{\mathrm{TP}}{\mathrm{TP}\;+\;\mathrm{FP}}\\ \mathrm{recall} & =\frac{\mathrm{TP}}{\mathrm{TP}\;+\;\mathrm{FN}}\\ F1 & =2\frac{\mathrm{precision}\cdot \mathrm{recall}}{\mathrm{precision}+\mathrm{recall}} \end{array}$$

In macro-averaging, all values are first calculated for each class and then averaged, giving each class the same importance for the final prediction independent of the number of samples for this class. We additionally report the performance of the Majority Baseline, i.e., a classifier that predicts the most common class (Progression) for each test sample as a practical lower bound.

Additionally, we test how well the model explanations (i.e., the as important identified words) align with domain knowledge. To this end, we ask the radiologist to indicate which words they used to label the reports (using the same subset as in Section 3). We then present the list of important words identified by the feature importance techniques and ask the radiologist to judge them for their relevancy to the decision. For example, the word “MRI” would be rated as a spurious feature, because all reports are written about MRI scans, and the word MRI gives no information about its class. The word “baseline”, however, can be considered valid for class 1 (diagnosing and (re)baseline). We summarise these findings into several word lists, namely *valid words* (one list per class), perhaps valid words (whose validity depends on the context, one list per class) and *wrong words* (spurious features).

We then determine for each word retrieved by the feature importance if they are (i) valid, (ii) perhaps valid, or (iii) invalid. A word could be invalid for two reasons. Either is contained in the list of “wrong words”, or it is in the “valid” or “perhaps valid” list of another class, but not in the list of the predicted class. From the feature importance methods, we obtain the five most important words with their importance score $s_{i}$ (cf. Section 4.2). We normalize these scores such that $\sum_{i}s_{i}=1$, manually correct misspellings and resolve synonyms. For each report, we then categorize the words as (i) valid, (ii) perhaps valid, and (iii) invalid using the manually curated list described above and sum the weights per category. Thus, we obtain an indication of how much the model relies on valid, perhaps valid, or invalid words. We summarise these results by averaging over all leave-one-out splits (i.e., averaging over all reports) to obtain a score for each BERT model.

## 5. Results

In the following, we report the performance of the models (cf. Section 5.1), the technical quality of the explanations and the verification of the explanations (cf. Section 5.2), and consequently of the models with domain knowledge (cf. Section 5.3). Finally, Section 5.4 reports on the results of the prospective study.

### 5.1. Technical Evaluation

The results of the leave-one-out evaluation are shown in Table 2. Overall, all tested combinations of feature representations and classifiers achieve high accuracy (≥0.79), outperforming a simple majority classifier (accuracy 0.65) by a large margin. The best classical model (TF-IDF + LR, F1 = 0.83) is on par with the best end-to-end deep learning approach (BERTje, F1 = 0.8), while the latter has better precision and recall trade-off. We find a huge difference in performance for the different BERT models, ranging from F1 = 0.63 for MedRobERTa.nl to F1 = 0.82 for BERTje. This can also be seen in Figure 2, which illustrates that only BERTje classified the majority of class 3 correctly. However, all three BERT models frequently classify classes 1 and 2 correctly.

### 5.2. Model Explanations

To validate the correctness and completeness of the explanation w.r.t. to the model, we report the model’s performance with only the important words, and without important words. Results are shown in Table 3. For both, SHAP and LIME, the reports can be classified with similar accuracy with only the five most important words than with the original report. Removing those words, however, also leads to similar accuracies. This means that the five words are important and sufficient for accurate classification, but without those words, there is enough information in other words in the report to classify them correctly. For IG, the accuracy with only the important words drops significantly (from the range of 0.90 to 0.70), while classification accuracy is higher without the important words. We conclude that IG fails to identify the words that are important for the classifier’s decision without additional context, leaving them at least partially in the reports “without important words”. Examples of (anonymized) reports with their respective highlighted words are shown in Appendix A.

### 5.3. Verification with Domain Knowledge

To analyze if the outcomes of the XAI techniques match with domain knowledge, we compare words identified as important by the feature importance techniques with lists formulated by a radiologist. The results are shown in Table 4. Spelling mistakes and synonyms are corrected manually. If a word from the XAI explanations has not been encountered by the radiologist, it is sorted in the “unknown” category. For all models, either SHAP or LIME is the most positive, as their results indicate that the models base their predictions for the most part on valid words. Specifically, (perhaps) valid words have—according to LIME—on average an impact of 0.54 on BERTje’s predictions, 0.52 for RobBERT and 0.41 for MedRoBERTa.nl. For SHAP this is 0.56, 0.51, and 0.38, respectively. This means that LIME and SHAP explanations align largely with domain knowledge.

### 5.4. Prospective Study

All additional MRI reports that were written in the hospital during the course of the work were subjected to a prospective study. Note that during the study time, only MRIs for diagnosis and progression were taken, thus, we do not have reports of the (rare) class “Monitoring” in our test set. We trained our best-performing end-to-end model (BERTje) on the full original dataset and evaluated it on the datasets with the new reports. We use the same preprocessing (including oversampling) and hyperparameters as reported before. Table 5 reports on the results of this study. We reported weighted recall, precision and F1, as there are no reports from class 3 (Monitoring) in the dataset. The accuracy (0.92) is comparable to the accuracy found in the retrospective study (0.90).

## 6. Conclusions and Future Work

In this work, we have investigated the task of classifying radiology reports from MS patients with three Dutch BERT models: two domain-generic models and one domain-specific model. BERTje outperforms the other tested BERT models and retains similar predictive performance in a prospective study. A simple logistic regression classifier in TF-IDF feature space, however, achieves comparable performance. We expect BERTje to benefit from a larger dataset, be able to exploit its distributed reasoning strategies and outperform the bag-of-words approaches. While BERTje (a domain-generic model) outperforming RobBERT (another domain-generic model) is not that notable, BERTje outperforming MedRoBERTa.nl (a domain-specific model) in the domain of the latter is quite notable. This implies that a domain-specific BERT model is not always better than a domain-generic model for a domain-specific task. Both RobBERT and MedRoBERTa.nl are RoBERTa-based, and BERTje (BERT-based) outperformed them both, so it could also be that BERT-based models are better suited for this specific task.

Assessing the most important words for the decision with the feature importance technique, we found (i) SHAP and LIME to produce correct features, which are (ii) largely in correspondence with domain knowledge.

Our reports were annotated by one radiologist because annotations from domain experts are expensive, training of radiologists with the annotation scheme requires additional time, and inter-annotator agreement on tasks involving natural language has been shown to be low. For instance, Bobicev and Sokolova report values between 0.5 and 0.7 for a sentiment classification task [47]. We observe an intra-annotator score of 0.6 (Cohen’s kappa) on all reports that were subjected to a second annotation (including the difficult ones), whereas for the random samples, we observe a score of 1.0. In order to obtain a solid ground truth, we repeated the annotation for reports that we differently labeled in previous steps (cf. Section 3). We acknowledge that in a reproduction study, slightly different ground-truth labels might emerge due to inter-annotator differences. However, we do not expect those differences to have a negative impact on the performance of the classifiers or the general results.

Labeling the reports is not necessarily unambiguous for radiologists. For cases where the model was frequently in disagreement with the original label, we observed a low intra-annotator agreement, whereas other cases had a high intra-annotator agreement. However, it is important to note that only a subsection of the dataset was reviewed and corrected. Therefore it is possible that a non-significant portion of the cases currently misclassified by BERTje cannot be classified by either the radiologist or the model due to their complexity or ambiguity. Alternatively, it is possible that the radiologist has misclassified some cases. The latter implies that the accuracy of BERTje might be underestimated, as more reports misclassified by BERTje could be misclassified by the radiologist instead.

## Figures and Tables

**Figure 1 diagnostics-13-01251-f001:**
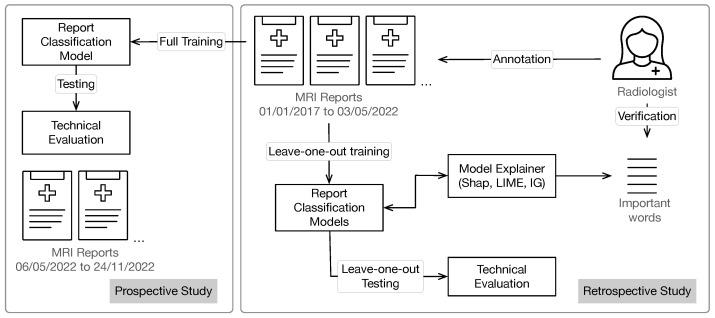
Overview of our study setup. In the retrospective study (right), we train and test models on data collected retrospectively. The models are evaluated with leave-one-out cross-validation. Three standard XAI feature importance techniques are applied to the trained models, and their resulting feature importance is verified by a domain expert. The explanations are additionally validated w.r.t. to their fidelity to the model they explain. The setup and results of the prospective study are reported in Section 5.4.

**Figure 2 diagnostics-13-01251-f002:**
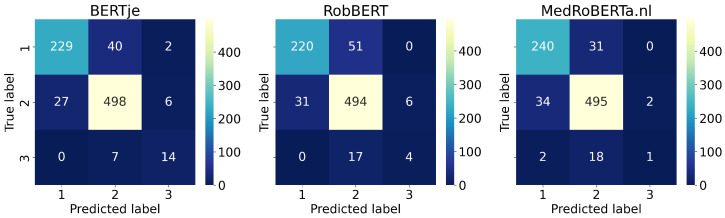
Confusion matrices for the three BERT models.

**Table 1 diagnostics-13-01251-t001:** MRI report classification scheme and report distribution.

Label	Description	Reports
Diagnosis	For initial diagnosis, or to create a (new) baseline	271
Progression	Progression or monitoring medication effect	531
Monitoring	Monitoring on secondary complications (mostly PML)	21

**Table 2 diagnostics-13-01251-t002:** Leave-one-out evaluation results reporting accuracy, macro-averaged precision, recall and F1. BERT-based model + LR/RF/SVM indicates that the classifier was trained on hidden states yielded by the encoder stack of the pre-trained BERT-based model. Best values marked bold.

	Accuracy	Precision	Recall	F1
**TF-IDF**				
+LR	0.89	0.92	0.78	**0.83**
+RF	0.86	**0.91**	0.60	0.63
+SVM	0.85	0.89	0.63	0.67
**doc2vec**				
+LR	0.82	0.63	0.65	0.64
+RF	0.78	0.68	0.57	0.61
+SVM	0.82	0.67	0.64	0.66
**BERTje**				
+LR	0.84	0.71	0.65	0.67
+RF	0.81	0.55	0.52	0.53
+SVM	0.82	0.65	0.65	0.65
end-to-end	**0.90**	0.81	**0.82**	0.82
**RobBERT**				
+LR	0.86	0.78	0.79	0.79
+RF	0.84	0.56	0.55	0.55
+SVM	0.79	0.63	0.81	0.66
end-to-end	0.87	0.72	0.64	0.67
**MedRoBERTa.nl**				
+LR	0.80	0.59	0.60	0.60
+RF	0.83	0.88	0.56	0.57
+SVM	0.80	0.62	0.71	0.64
end-to-end	0.89	0.70	0.62	0.63
**Majority Baseline**	0.65	0.22	0.33	0.26

**Table 3 diagnostics-13-01251-t003:** Accuracy with reduced reports, based on feature importance methods. Accuracy of the majority classifier is 0.65.

BERTje	SHAP	LIME	IG
full report	no selection: 0.90		
only important words	0.89	0.76	0.69
without important words	0.87	0.75	0.73
**RobBERT**	**SHAP**	**LIME**	**IG**
full report	no selection: 0.87		
only important words	0.87	0.80	0.68
without important words	0.87	0.80	0.73
**MedRoBERTa.nl**	**SHAP**	**LIME**	**IG**
full report	no selection: 0.89		
only important words	0.90	0.83	0.74
without important words	0.88	0.83	0.80

**Table 4 diagnostics-13-01251-t004:** Comparison between important words as identified by XAI methods and domain knowledge (manually constructed lists). “Unknown”: word not in manual lists, thus unable to determine which category it should fall in.

BERTje	SHAP	LIME	IG
valid	0.53 ± 0.26	0.47 ± 0.29	0.22 ± 0.18
perhaps valid	0.03 ± 0.08	0.07 ± 0.13	0.06 ± 0.11
unknown	0.35 ± 0.27	0.30 ± 0.28	0.32 ± 0.21
invalid: other	0.04 ± 0.08	0.10 ± 0.17	0.08 ± 0.13
invalid: wrong	0.05 ± 0.09	0.06 ± 0.11	0.32 ± 0.17
**RobBERT**	**SHAP**	**LIME**	**IG**
valid	0.49 ± 0.26	0.47 ± 0.26	0.26 ± 0.24
perhaps valid	0.02 ± 0.07	0.05 ± 0.10	0.03 ± 0.09
unknown	0.33 ± 0.28	0.27 ± 0.25	0.62 ± 0.26
invalid: other	0.11 ± 0.16	0.15 ± 0.16	0.04 ± 0.10
invalid: wrong	0.04 ± 0.09	0.07 ± 0.12	0.05 ± 0.10
**MedRoBERTa.nl**	**SHAP**	**LIME**	**IG**
valid	0.36 ± 0.25	0.37 ± 0.25	0.23 ± 0.20
perhaps valid	0.02 ± 0.07	0.03 ± 0.06	0.01 ± 0.05
unknown	0.46 ± 0.28	0.41 ± 0.26	0.53 ± 0.28
invalid: other	0.08 ± 0.13	0.12 ± 0.16	0.12 ± 0.18
invalid: wrong	0.07 ± 0.11	0.07 ± 0.13	0.11 ± 0.15

**Table 5 diagnostics-13-01251-t005:** Overview of the prospective study using a finetuned BERTje: number of reports in each dataset and the results on the new dataset. Reporting accuracy and weighted precision, recall and F1.

Dataset	Set Type	Type of Reports		
		Diagnosis	Progression	Monitoring
	Train (Original)	271	531	21
	Test (New)	19	101	0
**Results** **(BERTje)**	**Metrics**			
	**Accuracy**	**Precision**	**Recall**	**F1-Score**
	0.92	0.95	0.92	0.93

## Data Availability

The code presented in this study is openly available in gitlab.com/MTRietberg/nlp-ms-mri together with the anonymized example reports from Appendix A.

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
