# Peer review of "Accurate and Reliable Classification of Unstructured Reports on Their Diagnostic Goal Using BERT Models"

_diagnostics, 2023, doi:10.3390/diagnostics13071251_

Round 1

Reviewer 1 Report

Manuscript entitled "Accurate and Reliable Classification of Unstructured Reports on Their Diagnostic Goal Using BERT Models" submitted by Max Tigo Rietberg et al. described the BERT models which can predict patient’s diagnosis, progression and risk probabilities and advise treatments, monitoring will be useful for patient population. Please address the comments below.

11)     In the abstract, please abbreviate the terms used such as XAI, BERT as it represents the summary of the story.

22)     In line 51, who is a domain expert considered to be?

33)     Does BERT transformer models can be used for other languages?

44)     In line 70, BERT was introduced and expanded, however, this needs to be done in the abstract and introduction section as well to catch readers attention.

55)     Please mention how BERT models can be compared with traditional models in the MRI field. Do specify the advantages clearly with comparison.

66)     In fig 1 of the retrospective study, why there is no verification of technical evaluation step? Please provide an explanation.

77)     In table 2, please define what F1 is. Also are the values in table 2 majority baseline can be subtracted from all BERTje and other leave -on-out evaluation results?

88)     Of the table 1 reports of progression, what is the majority of BERT model predictions looks to be and how well they similar to each within the progression group? Similarly, describe for diagnosis and monitoring groups as well.

99)     Are there any drawbacks of BERT models? Please mention them in the manuscript.

110)  If the MRI scans are used to convert BERT model, please include an MRI image of a MS patient, convert and provide a rationale how a radiologist can read or predict and suggest a monitoring instruction for a MS patient. As a practical point of view, the model needs to be demonstrated well enough to predict and annotate the results.

Please address these comments as a major revision.

Reviewer 2 Report

This study investigated the task of classifying 826 radiology reports using Explainable AI (XAI) from 360 different multiple sclerosis patients with three Dutch BERT models: two domain-generic models and one domain specific model from the MRI scan.

The manuscript is well written. However, it can be further improved.

There are only few comments here:

Page 4, line 137: The authors use a dataset of 826 reports for MRI scans. However, in Table 1, there are only 823 reports (271 + 531 + 21). Three reports are missing. Is this related to statement in Page 5, line 185 because of the Exclusion?

Page 4, line 146: The dataset contains reports from nearly 5.5 years (2017-01-01 up to and including 2022-05-03). The use of 5.5 years is inaccurate since there are 12 months in a year. Better to use 5 years 4 months or over 5 years.

How many Radiologists involved in this study? Any error study was performed to rule out human error, especially from the Radiologist?

Page 8, line 287: Why there is no reports from the class 3 (Monitoring) in the data set?

Please be consistent of the use of word data set vs dataset.

Reviewer 3 Report

The authors have presented an interesting method to classifying radiology reports from MS patients. They have used three Dutch BERT models: two domain-generic models and one domain- 291 specific model for this study. Authors have demonstrated the performance of other related tools and has shown that BERTje outperforms the other tested BERT models. Comparison of deep learning models with classical approaches such as Support Vector Machine, Random Forest model and logistic regression classifiers are shown in this paper. Results shows that this approach utilizing NLP and XAI techniques are useful in classifying MS reports and determines the most reliable model. Feature importance using IG, LIME and SHAP XAI techniques, quantifies the impact of specific features on the model’s predictions. For the prospective study, authors have used the best performing BERTje evaluation method and shown good performance results with the new dataset.

 Some of the minor suggestions that can be addressed in this manuscript are the following:

1.     Abbreviations used in the manuscript can be made consistent throughout the paper. For example: TF-IDF is used as TFIDF in some parts of the paper.

2.     As reported in the retrospective study, is there any class imbalance of data observed and handled in the prospective study?

Round 2

Reviewer 1 Report

R1.C1 In the abstract, please abbreviate the terms used such as XAI, BERT as it represents the summary of the story. We expanded the abbreviations as suggested.

R1.C2 In line 51, who is a domain expert considered to be? We added the clarification in the text that the domain expert is a board certified radiologist specialising in neurology.

R1.C3 Does BERT transformer models can be used for other languages? In general the BERT model architecture is language agnostic, and there exists monolingual and multilingual models. Based on preliminary research on multilingual BERT models we used a monolingual BERT model in our paper and explained the reasons (multilingual models do not outperform monolingual ones) in the related work section.

R1.C4 In line 70, BERT was introduced and expanded, however, this needs to be done in the abstract and introduction section as well to catch readers attention. We added the clarification sentence on BERT to the introduction as follows: "We compare conventional NLP models against Bidirectional Encoder Representations from Transformers (BERT) models. BERT is a machine learning framework for NLP introduced by Devlin et al., composed of Transformer encoder layers and trained to learn latent representations of words and sentences."

R1.C5 Please mention how BERT models can be compared with traditional models in the MRI field. Do specify the advantages clearly with comparison. BERT models are standard models for natural language processing. We base our classification of MRI images solely based on the reports associated with the MRI images. We clarified this in the papers as outlined in comment R1.C11

R1.C6 In fig 1 of the retrospective study, why there is no verification of technical evaluation step? Please provide an explanation. Figure 1 shows that we performed a technical evaluation in the retrospective study using leave-on-out cross-validation (Results are shown in Table 2). Our best model in the retrospective study was evaluated on prospective data. The prospective data has a manual ground-truth assigned, and thus, we performed a technical evaluation for the prospective study (denoted as "Testing" in Figure 1). Results of this evaluation are shown in Table 5. We adapted Figure 1 to clarify the technical evaluation in the prospective study and extended the caption of the figure to show where the study and its results are described.

R1.C7 In table 2, please define what F1 is. We added a definition of our evaluation metrics at the beginning of section 4.3 as follows: “Accuracy, precision, recall and F1 can be defined in terms of the number of true positives (TP), true negatives (TN), false positives (FP) and false negatives (FN). Accuracy is the ratio of true positives and all predictions. Precision is the ratio of TP and all samples that were predicted as true. Recall is the ratio of TP and all samples that should have been predicted as true. F1 is the harmonic mean of precision and recall: ???????? = ?? ??+??+??+?? ????????? = ?? ??+?? ?????? = ?? ??+?? ?1 = 2 ?????????·????????? ?????????+?????? In macro-averaging, all values are first calculated for each class and then averaged, giving each class the same importance for the final prediction independent of the number of samples for this class.”

R1.C8 Also are the values in table 2 majority baseline can be subtracted from all BERTje and other leave -on-out evaluation results? The majority baseline reported in Table 2 represents a practical lower bound of performance metrics. A naive classifier always assigning the majority class (class Progression) would achieve these performance values. As such they stand on their own for comparison and cannot be subtracted from the other values. We added "We additionally report the performance of the Majority Baseline, i.e., a classifier that predicts the most common class (Progression) for each test sample as a practical lower bound." as clarification in Section 4.3.

R1.C9 Of the table 1 reports of progression, what is the majority of BERT model predictions looks to be and how well they similar to each within the progression group? Similarly, describe for diagnosis and monitoring groups as well. We added the confusion matrices for the BERT models in Figure 2, with the following observation in Section 5.1: “We find a huge difference in performance for the different BERT models, ranging from F1=0.63 for MedRobERTa.nl to F1=0.82 for BERTje. This can also be seen in Figure 2, which illustrates that only BERTje classified the majority of class 3 correctly. However, all three BERT models frequently classify class 1 and 2 correctly".

R1.C10 Are there any drawbacks of BERT models? Please mention them in the manuscript. We added the limitations of BERT in the related work section as follows: "BERT's primary limitations are its significant model size that makes it computationally demanding to train and apply, as well as its incapability to handle non-English corpora."

R1.C11 If the MRI scans are used to convert BERT model, please include an MRI image of a MS patient, convert and provide a rationale how a radiologist can read or predict and suggest a monitoring instruction for a MS patient. As a practical point of view, the model needs to be demonstrated well enough to predict and annotate the results. We base our prediction solely on clinical reports written by a radiologist for the MRI scans. We do not use MRI imaging data. The reports of the MRI scans were categorised by a radiologist into one of 3 classes (cf. Table 1). This was done in accordance with the MAGNIMS (Magnetic Resonance Imaging in MS) standards. We clarified this in the introduction as follows: "Using NLP models, we extract the reasons for taking the MRI, quantifying the model's ability to classify MRI reports from MS patients. It's worth noting that we do not classify the MRI scans directly, but rather the accompanying reports." and section 3 as follows "Each report was manually classified by a radiologist into one of three classes: 1) Diagnosis, 2) Progression, or 3) Monitoring. This classification was performed in accordance with the MAGNIMS (Magnetic Resonance Imaging in MS) criteria.".

Thank you for the responses. Please accept the revised version.